# Sweet Orange: Evolution, Characterization, Varieties, and Breeding Perspectives

**Sebastiano Seminara [1,†], Stefania Bennici [1,†], Mario Di Guardo [1], Marco Caruso [2], Alessandra Gentile [1,3], Stefano La Malfa [1,\*] and Gaetano Distefano [1]**

1  Department of Agriculture, Food and Environment (Di3A), University of Catania, via Valdisavoia 5, 95123 Catania, Italy
2  Council for Agricultural Research and Economics (CREA)—Research Centre for Olive, Fruit and Citrus Crops, Corso Savoia 190, 95024 Acireale, Italy
3  College of Horticulture and Landscape, Hunan Agricultural University, Changsha 410128, China
\*  Correspondence: slamalfa@unict.it
†  These authors contributed equally to this work.

**Abstract:** Among Citrus species, the sweet orange (*Citrus sinensis* (L.) Osbeck) is the most important in terms of production volumes and cultivated areas. Oranges are particularly appreciated for the organoleptic characteristics and the high nutraceutical value of the fruits (thanks especially to their high content of antioxidants). Recent advances in citrus genetic and genomic resources, such as the release of the reference genomes of several sweet orange cultivars, have contributed to (i) understanding the diversification of *C. sinensis* and its relation with other citrus species, (ii) assessing the molecular mechanisms underlying traits of interest, (iii) identifying and characterizing the candidate genes responsible for important phenotypic traits, and (iv) developing biotechnological methods to incorporate these traits into different citrus genotypes. It has been clarified that all the genetic diversity within the sweet orange species was derived from subsequent mutations starting from a single ancestor and was derived from complex cycles of hybridization and backcrossing between the mandarin (*Citrus reticulata* Blanco) and the pummelo (*Citrus maxima* (Burm.) Merr.). This paper provides an overview of the varietal panorama together with a description of the main driving forces in present and future sweet orange breeding. In fact, for the sweet orange, as well as for other citrus species, the release of novel varieties with improved characteristics is being pursued thanks to the employment of conventional and/or innovative (molecular-based) methods. The state of the art methods together with the innovations in genomics and biotechnological tools leading to the so-called new plant breeding technologies were also reviewed and discussed.

**Keywords:** *Citrus sinensis*; biodiversity; fruit quality; clonal selection; NPBTs

## 1. Introduction

The sweet orange (*Citrus sinensis* (L.) Osbeck) is the most important species among those belonging to the Citrus genus, representing about 50% of global citrus production. Worldwide, this species is grown in many tropical and subtropical regions in areas approximately located between the latitudes of 35° north and 35° south . According to FAO, the sweet orange is cultivated worldwide on more than 3.8 million hectares of land with a corresponding production of 75.5 million tons (FAOSTAT 2020). Brazil, India, and China are the main producing countries (with 16.7, 9.8, and 7.6 million tons produced, respectively) followed by the United States of America (with 4.8 million tons) (FAOSTAT 2020). In recent years, sweet orange production in Brazil and the United States of America has significantly declined due to the spread of the bacterial disease Huanglongbing (HLB). The disease, caused by the phloem-limited Gram-negative bacterium "*Candidatus*

Liberibacter spp." [1], is considered the most threatening pest to citrus plants. C. Liberibacter's origin and first diffusion are uncertain although it has been observed in China for over a century [2]. The HLB disease was found in the American continent less than 20 years ago: it was first reported in Sao Paulo, Brazil, in 2004 [3] and then spread to Florida in 2005 [4], causing significant economic losses. No cases of HLB are currently being reported in the Mediterranean countries even though one of the two known vectors of the disease, *Trioza erytreae*, was detected in Portugal and Spain and although *Diaphorina citri* was recently discovered in Israel [5].

In Europe, sweet orange production accounts for more than 6.4 million tons of sweet oranges, with Spain and Italy representing the main producers with 3.3 (6th world producer) and 1.8 million tons (10th world producer) produced, respectively (FAOSTAT 2020). According to FAO, in 2020, 84,160 hectares of land was cultivated in Italy, mainly for navel and blood orange production, with a yield of more than 21 tons/ha. The fruits are often consumed fresh even though a considerable part is processed to produce orange juice, which is supplied worldwide, mainly from Brazil and Florida [6].

The oldest documents reporting the existence of orange fruits can be dated back to ancient China. Specifically, the first reference can be found in the book 'Tribute of Yu', dedicated to the Chinese emperor Ya Tu (who ruled from 2205 to 2197 B.C.), in which the following statement is reported: "*The baskets were filled with woven ornamented silks. The bundle contained small oranges and pummeloes.*" [7].

Recent phylogenetic studies report that the sweet orange is derived from complex cycles of hybridization and backcrossing in which one or more intermediate individuals are still unknown, having the mandarin and the pummelo as its founders (Figure 1) [8,9]. The identification of true citrus species provided new information about the phylogeny, the origins, the evolution, and the spread of the most important citrus species and varieties. Based on new evidence from a whole-genome analysis, a recent phylogenomic classification proposed a new taxonomical name for the sweet orange, *C. × aurantium* var. *sinensis* L., since both the sour orange and the sweet orange are derived species sharing the same ancestors [10]. Hundreds (or maybe thousands) of years of cultivation have subsequently generated a multitude of cultivars selected by growers, horticulturists, and breeders for their special characteristics, and they subsequently underwent clonal propagation, mainly through grafting. In this regard, the diversification characterizing the vast array of sweet orange varieties is one of the most evident examples of the role of somatic mutations in determining intraspecific diversification. Due to this, hundreds of cultivated clones have been selected, differing due to the peculiar characteristics of the plant or of the fruit itself (e.g., fruit size, ripening period, peel and flesh color, presence of seeds, and acidity) [11–13].

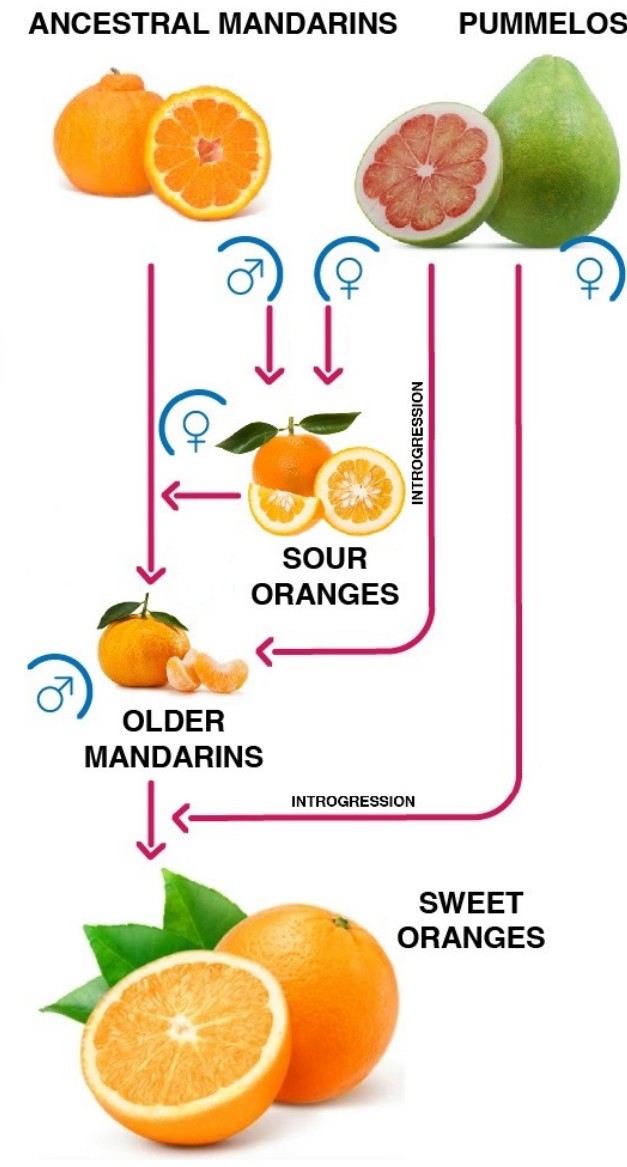

**Figure 1.** Sweet orange phylogeny[ ♂ male, ♀ female, red arrow: phylogenetic relationship].

## 2. A Single Hybrid Ancestor, Different Fruit Typologies

Despite the sweet orange being derived from a single unknown ancestor, the currently known sweet orange varieties show a wide variability in terms of fruit characteristics (Figure 2), especially those of color, taste, yield, maturity date (Figure 3), and many other horticulturally important traits. Such a wide variability is the result of the subsequent field selection, propagation, and diffusion of selected varieties in different cultivation areas throughout the years.

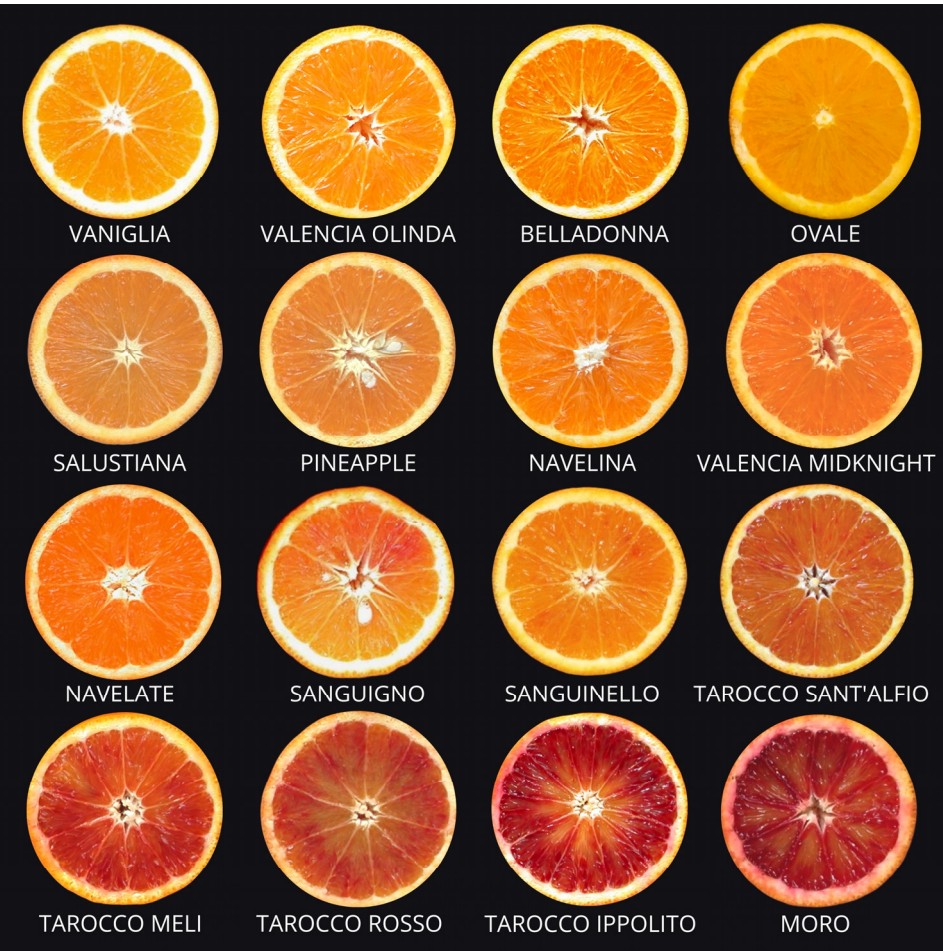

**Figure 2.** Different types of sweet orange fruits.

Many different selection criteria, mostly derived from local perceptions and the personal evaluation of the quality traits of the fruit (hesperidium), have been considered. The hesperidium is a particular kind of berry characterized by the presence of multiple segments (endocarp) surrounding a central axis (columella) and containing vesicles, structures arising from the inner part of the ovary. These structures, growing into locular cavities, give rise to elongated sacs inside which the watery juice is accumulated. The peculiar hesperidium traits refer to the presence of a multilayer peel (flavedo and albedo) harboring glands containing the essential oils of the endocarp and of the central axis. Additionally, the fruit can exhibit peculiar structures either in the proximal part of the fruit (forming a neck) or in its distal part, where a "navel" can be found. This represents a secondary fruit growing inside the main one [14]. The sweet orange varieties are commonly divided into four subgroups:

(1) Common oranges, comprising many varieties that are different in origin, use, presence of seeds, and ripening time;
(2) Navel oranges, in which a secondary fruitlet (navel), which develops within the primary fruit, occurs;
(3) Pigmented or blood oranges, which accumulate moderate to high levels of anthocyanins in the flavedo and/or flesh during ripening;
(4) Sugar or acidless oranges, which have very low acidity in the pulp, a flat flavor, and a consequent low diffusion and commercial importance.

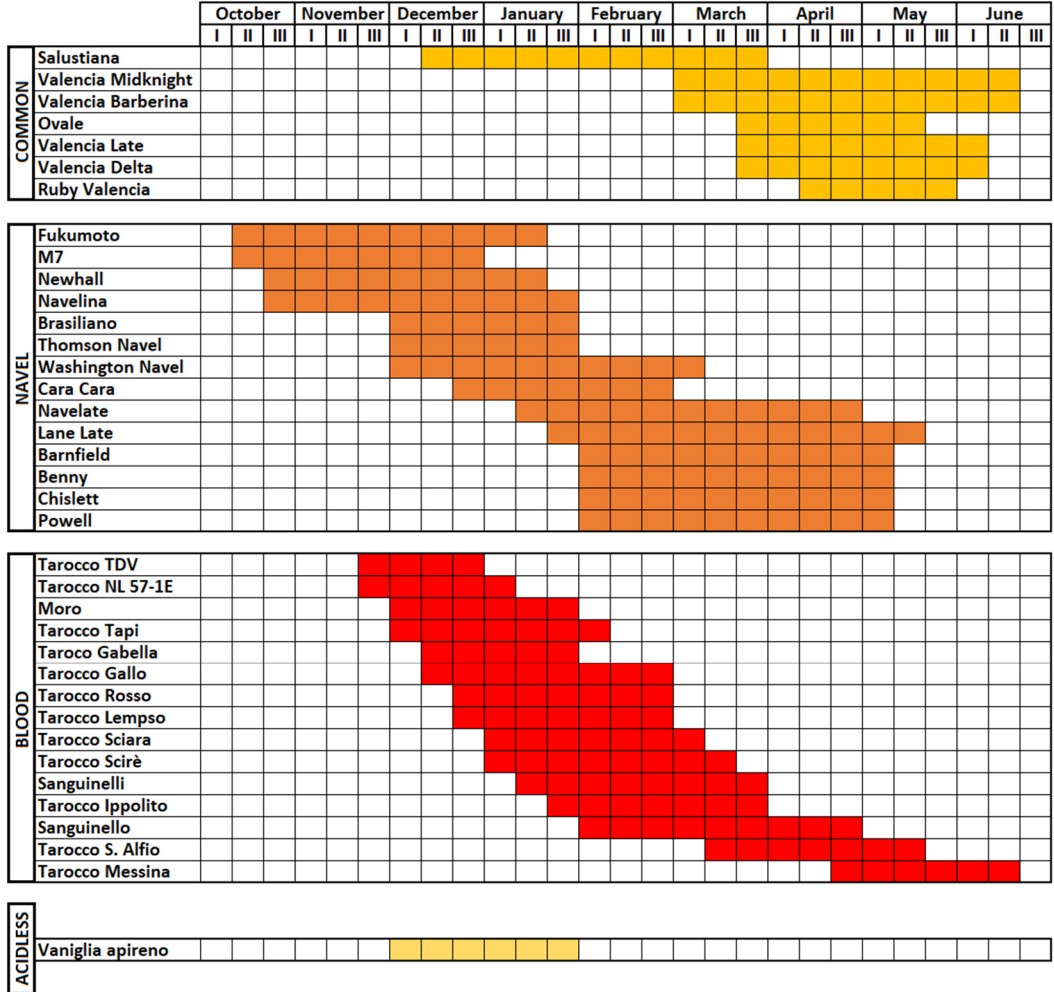

**Figure 3.** Ripening period of some of the most diffuse varieties in the Mediterranean area (orange color:.sweet orange common subgroup; dark orange: navel subgroup; red color: blood subgroup; pale orange: acidless subgroup).Their variability strongly depends on their cultivation environments and rootstock influence.

## 2.1. Common Oranges

This sweet orange subgroup is the most widely cultivated and marketed in the world. Common oranges encompass a wide number of varieties, all showing "blond" flesh, but they are rather different with respect to many other traits of interest (e.g., seed presence, fruit quality, yield, ripening time, and resistance to stress). Many of these cultivars are suitable for industrial processing due to their high juice yield and low content of limonin, a terpenoid responsible for juice bitterness when present in high quantities [15]. The most widespread varietal group is the "Valencia", which was probably derived from a nucellar seedling of the "Selecta" variety and was found in Portugal in the 19th century. Nowadays, Valencia oranges are among the most widely cultivated varieties worldwide, with fruits used both for fresh consumption and for processing [16]. The success of the "Valencia" orange has been largely determined by its adaptability to different climatic conditions, its high productivity, and its good fruit conservation both on the plant and during postharvest, and the fruits are usually seedless. The "Valencia" varietal group consists of mainly late-maturing clones, such as the "Olinda" and the "Campbell" as well as the more recent "Barberina", "Delta", and "Midknight" [17]. Growing attention is also being paid by growers to the clones "Rhode Red", which is characterized by an intense coloration of the peel and the flesh [18,19], and "Ruby Valencia", which is characterized by a pink coloration of the flesh due to a high accumulation of lycopene [20,21]. Recently, several early-

("EV1", "EV2", and "Valquarius") and late-ripening clones (clones of the "OLL" series—Orie Lee Late) have been released in Florida and are more and more diffusely cultivated [22]. Other popular common oranges are the "Pineapple" and "Hamlin", mainly used for industry; the "Pera", selected and diffused throughout Brazil due to its exceptional productivity; and the "Salustiana", diffused throughout Spain and Latin America and found earlier than the "Valencia". The "Shamouti", although its diffusion is limited to Israel, is another common variety that is widely cultivated, and it is particularly appreciated for its easy peelability and excellent fruit flavor [23]. In Italy, among the local most interesting cultivars, the "Ovale" (or the "Calabrese") is an old cultivar that is still widespread in some pedoclimatic niches; it is mainly found in coastal niches because it fears drops in temperature [24]. It is appreciated for its late ripening, excellent fruit firmness, and resistance to preharvest fruit drop [25]. The original clone is derived from a chimeral mutation of the "Biondo commune" (one of the oldest Italian varieties) and, therefore, presents a certain instability, causing frequent ancestral returns of branches that produce seeded orange "Biondo-type" fruits (Figure 4) [26]. Its seed presence has been overcome through the selection of nucellar lines [27].

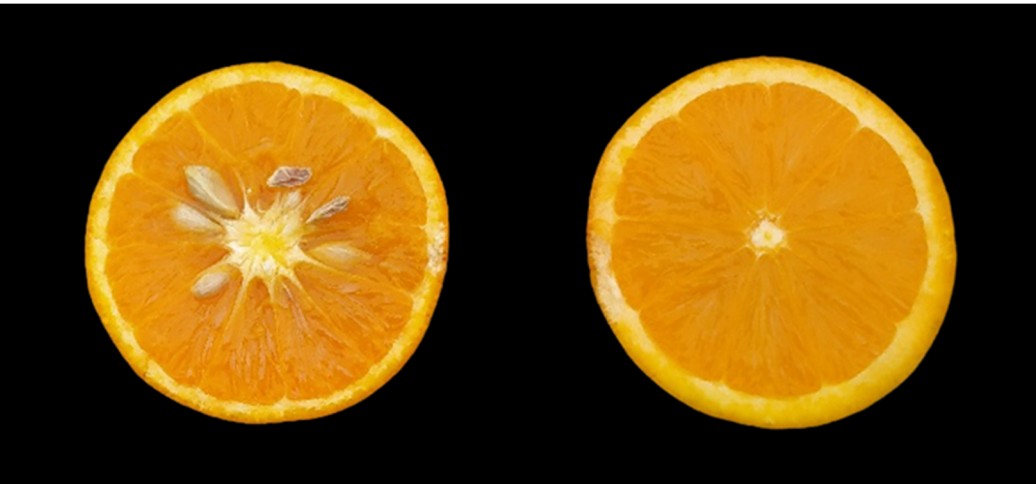

**Figure 4.** Fruit of "Ovale" orange (**right**) compared to a seedy fruit originated from a chimeric branch of the old "Ovale" line (**left**).

### 2.2. Navel Oranges

The common feature of the cultivars belonging to this group is the occurrence of syncarpy, i.e., the presence of a secondary fruitlet (called a "navel"). This phenomenon causes the extroflexion of the distal part of the fruit and the lack of regular scarring at the detachment point of the style from the ovary [26]. The exact origin of the "Navel" orange is unknown, but it is believed that all the different clonal selections of the "Navel" currently cultivated worldwide are derived from the "Washington Navel", a bud mutation of the "Selecta" cultivar found in the early 1800s in Bahia and Brazil and imported by the U.S. Department of Agriculture (USDA) in Washington DC in 1870. Subsequently, the "Washington Navel" spread rapidly to the other citrus-growing areas of the world. "Navel" fruits are characterized by high levels of limonin, making them suitable for fresh consumption and not for juice production. The high mutation rate of navel oranges contributed significantly to their diffusion, with many varieties being selected and propagated around the world. New accessions can sometime be hardly distinguishable from the original clone, except for during the ripening period, which, in some areas, can differ even by 6 months or more. The most popular early clones are the "Fukumoto", "Newhall", and "Navelina" [28]. Among the intermediate ripening clones, the "Washington Navel" remains the most widespread. Interest in late clones (the "Chislett", "Powell", "Lane Late", "Barnfield", and "Benny") has also increased in recent decades [29]. Clones with pink

flesh pigmentation are also gaining popularity. Their reddish-pink color is caused by lycopene accumulation. The most diffuse pigmented variety is the "Cara Cara", a probable mutation of the "Washington Navel" from Venezuela that has been propagated in Florida since 1990 [30]. More recently, other clones with the same pigmentation type, the "Kirkwood Red" and "Red Lina", were found in South Africa [23].

*2.3. Pigmented or Blood Oranges*

Pigmented oranges are characterized by the presence of red pigmentation in the pulp and sometimes also in the peel, which is determined through the synthesis of anthocyanins, water-soluble compounds belonging to the flavonoid group [31]. The varieties belonging to this subgroup are mainly widespread in South Italy, where most of the cultivated varieties originated, although it is very likely that the first pigmented ancestral variety was selected in China or Southeast Asia [32]. The varieties of pigmented oranges were grouped by Chapot in his description and classification work [33] into three groups based on their Mediterranean areas of distribution:

– Ordinary blood oranges: these comprise the three varieties selected and spread in Sicily (the "Sanguinello", "Moro", and "Tarocco") and the "Maltese Sanguigno", which is of unknown origin but was probably selected in Malta and subsequently spread throughout several North African areas [23];

– Doble Fina varieties: these comprise a Spanish group originated from the "Doble Fina" variety from which several accessions were selected. The "Sanguinelli" variety, not to be confused with the Sicilian "Sanguinello", belongs to this group. It was discovered in 1929 from a bud mutation of the "Doble Fina" in Castellón (Spain) and became widely popular due to its significantly higher levels of flesh and skin pigmentation compared to the original clone [34];

– "Shamouti" or "Palestine Jaffa" blood oranges: these comprise a small group, including the "Shamouti Maouardi" and "Maouardi Beladi" varieties, that are all accessions with similar characteristics to the blond "Shamouti", except for fruit pigmentation [35].

In citrus fruits, most cultivated varieties do not accumulate anthocyanins, which is related to the loss-of-function mutations in the *Ruby* gene cluster involved in the activation and deactivation of anthocyanin biosynthesis [36,37]. Along with *Ruby*, the *Noemi* gene (a Myc-like gene) controls anthocyanin pigmentation and, to a lesser extent, acidity [38]. However, despite their common genetic basis, blood orange selections display a wide range of anthocyanin pigmentation levels both in the pulp and/or the peel. The activation of the *Ruby* gene has been demonstrated to be cold-dependent [39], even if additional unknown molecular mechanisms are involved in anthocyanin biosynthesis and accumulation. Despite their high economic value, their diffusion is hampered by their low adaptability to different environmental conditions, especially when it concerns the synthesis of anthocyanins. Even varieties such as the "Moro" with a high pigmentation potential are highly dependent on the climatic conditions during the ripening period for the complete development of the typical coloration. "Moro" pigmentation seems to require less cold accumulation than the rest of the blood oranges since it starts to synthesize anthocyanins in mid-November in the typical growing conditions of Southeast Sicily about 2–3 weeks earlier than the commercial "Tarocco" selections (Marco Caruso, personal communication). During postharvest, storing the fruit at a low temperature can be a useful strategy to improve the pigmentation degree [40,41]. The dependence of anthocyanin accumulation on the growing environment has, therefore, limited the prevalence of blood orange cultivation in Italy; in particular, this has occurred in the areas of Sicily close to Mount Etna due to particularly suitable conditions [42]. Nevertheless, the interest of other citrus-growing countries, such as China, Spain, California, South Africa, and Australia, is increasing, although unsuccessful past experiences impose caution in their choice of cultivation environment. Suboptimal conditions during fruit ripening in areas of the American

continent where the world's largest production extensions are located have in fact resulted in fruits with generally weak or absent coloring. Additionally, in the citrus exporter countries of the southern hemisphere, the blood orange varieties become soft and drop before reaching their optimal internal fruit coloration [35]. This limits their propagation in these regions, creating the need to breed blood oranges with those traits as well.

Compared to the others, the greatest spread of the "Tarocco", a spontaneous mutant of the "Sanguinello" found in the early 1900s in the Syracuse Province, Sicily [25], resulted in the identification of a high number of vegetative mutations, showing high variability in terms of the ripening period (from December to May) and pigmentation degrees. Within this wide variability, the most recent discovery is represented by the "Tarocco Vigo", a spontaneous mutation selected from an old line of the "Tarocco" characterized by earlier and significantly higher anthocyanin accumulation (Figure 5).

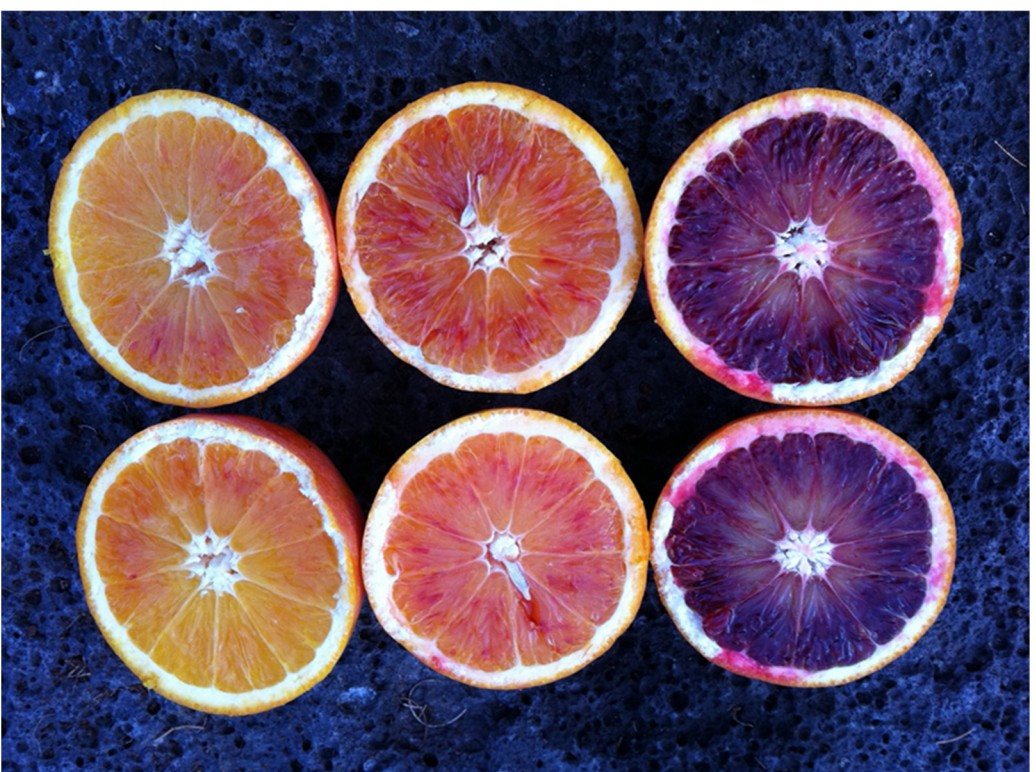

**Figure 5.** Fruit of "Tarocco Vigo" (**right**) compared to "Tarocco Comune" (**middle**) and "Tarocco dal Muso" (**left**) in mid-January (photo courtesy of Mr. Corrado Vigo).

Furthermore, in the United States, another two interesting pigmented sweet orange accessions were recently described: the "Valencia Smith Red" and the "Shahani", both mutants of the unpigmented varieties. The first was selected in California and is characterized by anthocyanin accumulation in the peel and flesh. Despite this area being historically poorly suited for blood orange production, this accession was described to be a good combination in terms of productivity, pigmentation, and taste [32]. The latter is the new "Navel" mutant called "Shahani", whose fruits are marked by evident anthocyanin pigmentation of the peel and flesh, few or no seeds, a smooth skin, and the characteristic fruitlet. The mutation was discovered by Mr. Frank Shahani in Southern California from an old "Washington Navel" plant and has not yet been propagated in cultivation because it is still under evaluation by the USDA in Riverside, California [43].

### 2.4. Sugar or Acidless Oranges

The "acidless" oranges are widespread in different Mediterranean areas under different names (the "Vaniglia" in Italy, the "Sucreña" in Spain, and the "Sukkari Mawardi"

in Tunisia). This small subgroup includes cultivars with fruits characterized by a very high sugar/acid ratio (80/100) caused by the almost total absence of acidity (to the order of 0.1%, about 1/10th of the value commonly found in oranges). These characteristics together with the absence of the typical "orangey" aroma give rise to a substantially flat flavor that has strongly limited their spread with minor exceptions in some Middle Eastern countries and, to a latter extent, in Spain and Portugal. There are only slight differences between these varieties, mainly related to their ripening period. Many accessions are the local names of the same variety [23].

## 3. Pomological Qualitative Traits

Sweet orange varieties are generally selected on the basis of traits such as fruit size and shape, rind and pulp color, flavor, and the absence of seeds. Furthermore, many other traits of the hesperidium are considered for the evaluation and selection of improved clones.

The traits of interest can vary according to the destination of the fruit (fresh or processed) [14]. If fruit quality traits such as juiciness and the TSS/acid ratio are of paramount importance, many others are important in determining the fruit value for specific use. In fact, for fruit devoted to industrial transformation, a high limonin content represents a detrimental factor for juice production. On the other hand, the external appearance of fruits is less important for industrial use, although different qualitative standards can be required for processing [44]. Furthermore, in the United States, some varieties (such as the "Valencia B9-65" and "Hamlin N13-32") have been specifically selected for their improved juice characteristics [45]. Citrus fresh fruit quality standards are largely dependent on the consumers' preferences and may change according to space and time [46,47]. Seedlessness is commonly an important and desirable fruit feature for fresh consumption [48,49]. The commercial maturity index of orange fruits is highly variable and depends on the variety, growing region, and target market. Nutritional and bioactive compounds undergo many changes during the ripening process. In any case, the sweet orange, as well as all other citrus fruits, is nonclimacteric and should be harvested when a minimum of internal maturity has been achieved (Table 1) [50].

The most widely used maturity indices that can be used to monitor the sweet orange ripening process are the juice content (%), total soluble solids (TSS; °Brix), TSS/acid ratio, and percentage of the fruit exhibiting typical coloration. Juiciness increases during maturation, reaching its maximum at full maturity and decreasing afterward [51]. Furthermore, the juice content may vary during fruit transportation. For this reason, the export of the fruits from countries of the Southern Hemisphere (Argentina, Uruguay, and South Africa) to other major markets (EU and USA) can be performed mainly on varieties characterized by a high juice content [52].

**Table 1.** Main maturity indices required in EU countries for sweet orange fresh fruits

|  | Minimum Juice Content (%) | Minimum Sugar/Acid Ratio |
|---|---|---|
| Blood oranges | 30 | 6.5:1 |
| Navel group | 33 | 6.5:1 |
| Other varieties | 35 | 6.5:1 |
| "Mosambi", "Sathgudi", and "Pacitan" (with more than one fifth of green color) | 33 | |
| Other varieties (with more than one fifth of green color) | 45 | |

Fruit color is variety-dependent. Generally, the green color of the peel may not exceed one fifth of the area of the fruit peel provided it satisfies the minimum requirements for the juice content. However, this value may be exceeded in the case of oranges produced in warm areas.

The differential sugar content of sweet orange fruits is represented by the °Brix percentage, expressed by the sugar content in g per 100 g of juice. The sugar/acidity ratio is considered the main maturity index of sweet orange fruits and is one of the main parameters of maturity and palatability. The TSS content is a widely used index whose level increases during maturity and helps in defining the optimal harvesting window. The TSS is composed of 80% of sugars (mainly fructose, glucose, and sucrose), 10% of acids (citric, malic, and oxalic acids), and 10% of nitrogenous compounds (i.e., amino acids). During maturation, the sugar content increases together with a decrease in organic acids, with citric acid as the main component (70–90%), followed by malic and oxalic acids [53].

Color is another vital attribute in sweet orange fruit quality, and it directly influences consumer perception and buying habits [54]. The peel color is correlated with the carotenoid composition and shows differences according to the varieties [55,56]. These differences can be measured using colorimetric parameters (CIELAB or Hunter L, a and b units). The a/b ratio or the CCI (Citrus Color Index, 1000×a/L×b) are the most widely used parameters [57]. During postharvest, the exogenous application of ethylene (degreening) can stimulate the coloration of the peel (but not the flesh). In general, the CCI requirements for degreening are between –5 and +3 [58].

*Effect of Environment and Agronomical Practices on Fruit Quality*

Together with the genetic background, environmental conditions strongly affect citrus growing and fruiting. Additionally, pedological conditions, the scion–rootstock combination, cultural practices, and even the tree age strongly affect the fruitification cycle.

One of the traits showing the highest environmental influence is the ripening period. As an example, the time between the blooming and harvesting of the "Valencia" orange lasts from 6 to 7 months in the low tropics to 14–16 months in Mediterranean-type climates [59]. Therefore, the same variety may exhibit significantly different fruit quality characteristics in regions of different climatic conditions. The fruit quality attributes which are particularly affected by climatic conditions include the juice content, citric acid content, °Brix/acidity ratio, juice pH, rind thickness, flesh percentage, fruit's shape index, and weight [60]. Color development is notably affected by climatic conditions such as light and temperature [61]. In warmer regions, characterized by low temperature excursion between night and day, citrus peel coloration is usually paler than it is in regions with greater day–night temperature fluctuations. It has long been recognized that the peel texture and adherence are also markedly affected by the temperature regime during the ripening period and thereafter [62].

Rootstocks can also influence growth to some extent; their main effects on the tree's characteristics are related to the growth habit and survival, yield, juice quality [16,63], ripening period [64,65], and ability of the tree to retain fruit [66]. Rootstocks affect primary internal fruit factors such as the juice content, color, soluble solids, acid concentrations, and their ratio, factors that basically define the internal quality of fruits because they are strongly related to taste [18,67,68] as well as the metabolic responses and antioxidant potential [69]. Additionally, external conditions such as fruit size and shape, rind thickness, color, and appearance are other critical marketing elements influenced by rootstock [70].

The new challenge for worldwide citriculture is, nowadays, represented by HLB [71]. Many studies showed that the rootstock does not affect the disease incidence since the trees on all rootstocks are susceptible to HLB. Nevertheless, tolerance to HLB is higher in trees grafted on some rootstock selections, and the use of a tolerant rootstock has been considered as an effective means to limit crop losses due to HLB. Among commercially available rootstocks, US-942 [72] appeared to have a clear advantage for commercial use under infected conditions, producing more fruit and having good fruit quality for a longer period [73,74]. Regarding the effect of scion cultivars, some HLB-tolerant clones or escape trees have been identified, but long-term field evaluations are still underway to confirm their tolerance.

## 4. Fruit Bioactive Compounds

Citrus flavor depends on a complex combination of soluble (organic acids, sugars, and flavonoids, which influence the taste) and volatile compounds (which influence the aroma) [75]. The taste is the result of the balance of the sweetness, bitterness, and sourness components. The sweet component is mainly due to three main carbohydrates, i.e., sucrose, fructose, and glucose [76]; the flavanones naringin and neohesperidine develop bitterness, while citric acid and malic acid are responsible for the sour taste [77]. Sweet oranges play an important role in the human diet as a functional food thanks to their wide range of bioactive compounds, such as polyphenols, carotenoids, and limonoids [78].

### 4.1. Primary Metabolites

The primary metabolites of sweet oranges consist mainly of sugars, organic acids, and lipids. Sugars are mono- and disaccharides, such as glucose, neohesperidose, and rutinose [79]. Citric acid is the main organic acid in sweet orange fruits together with other less abundant acids, such as malic, tartaric, and oxalic acids [80]. Malic acid is more abundant in unripe fruits and contributes to their sour taste. Sweet orange fruits are also a good source of other antioxidant compounds, such as ascorbic acid (Vitamin C), with differences observed among different varieties (Table 2), cultural practices, stages of ripening, climates, processing factors, etc. [81]. In human health, vitamin C is reported to play an important role in preserving connective tissues and in bone formation [82]. It is also involved in other metabolic pathways, such as B vitamin and folic acid biosynthesis, the conversion of cholesterol to bile acids, and many others [83]. Vitamin C also has antioxidant properties preserving cells from oxidative stress [84]. The "Tarocco" and "Sanguinelli", the most common and widespread blood orange varieties in the Mediterranean countries, were reported to be a good source of ascorbic acid (100 g of the edible portion consist of 70% of the recommended dietary allowance) [85]. Breeding varieties with similar characteristics are currently highly considered in order to produce fruits and derived products with nutraceutical properties, to be used for their antioxidant properties, or for their contribution of vitamin C [86].

Furthermore, sweet orange seeds are a good source of oils. Sweet orange seeds mainly contain linoleic, oleic, and palmitic acids as fatty acids, and a higher overall amount of unsaturated fatty acids versus their saturated counterparts has been identified [87]. Phytosterols are also found in citrus seed oil and have received attention for their antioxidant [88] and anticholesterol activities [89].

**Table 2.** Comparison of antioxidant activity of fresh juices of pigmented sweet orange varieties.

| Compound Class | Compound Name | Sanguinello | Moro | Tarocco |
|---|---|---|---|---|
| **Hydroxycinnamic acids** | Chlorogenic acid (mg/L) | 1.40 ± 0.26 | 4.80 ± 4.15 | 5.45 ± 5.49 |
| | p-Coumaric acid (mg/L) | 1.40 ± 0.26 | 1.42 ± 1.91 | 1.61 ± 1.31 |
| | Ferulic + sinapic acid (mg/L) | 5.91 ± 1.11 | 4.57 ± 3.76 | 3.68 ± 1.55 |
| **Flavanone glycosides** | Narirutin (mg/L) | 17.22 ± 3.24 | 18.25 ± 2.79 | 14.17 ± 2.25 |
| | Hesperidin (mg/L) | 189.20 ± 35.59 | 174.28 ± 13.13 | 217.77 ± 48.19 |
| | Didymin (mg/L) | 6.60 ± 1.24 | 6.80 ± 1.05 | 5.54 ± 0.38 |
| **Anthocyanidin glycosides** | Cyanidin-3-glucoside (mg/L) | 5.18 ± 3.99 | 46.30 ± 19.88 | 10.33 ± 11.63 |
| | Cyanidin-3-(6 "-malonyl)-glucoside (mg/L) | 7.33 ± 1.55 | 53.98 ± 1.06 | 25.08 ± 6.36 |
| | **Vitamin C** (ascorbic acid (mM)) | 3.27 ± 0.27 | 3.32 ± 0.25 | 3.11 ± 0.07 |
| **Compound Class** | **Compound Name** | **W. Navel** | **Valencia** | **Ovale** |

| | | | | |
|---|---|---|---|---|
| **Hydroxycinnamic acids** | Chlorogenic acid (mg/L) | 1.94 ± 0.12 | 1.84 ± 0.03 | 2.32 ± 0.59 |
| | p-Coumaric acid (mg/L) | 0.16 ± 0.04 | 0.15 ± 0.04 | 0.47 ± 0.30 |
| | Ferulic + sinapic acid (mg/L) | 0.76 ± 0.01 | 1.11 ± 0.03 | 1.31 ± 0.61 |
| **Flavanone glyco­sides** | Narirutin (mg/L) | 5.96 ± 0.21 | 4.57 ± 0.96 | 10.17 ± 4.49 |
| | Hesperidin (mg/L) | 100.75 ± 10.35 | 52.05 ± 13.22 | 121.73 ± 27.57 |
| | Didymin (mg/L) | 2.80 ± 0.01 | 1.86 ± 0.38 | 4.80 ± 2.20 |
| **Anthocyanidin gly­cosides** | Cyanidin-3-glucoside (mg/L) | nd | nd | nd |
| | Cyanidin-3-(6 "-malonyl)-glucoside (mg/L) | nd | nd | nd |
| | **Vitamin C** (ascorbic acid (mM)) | 2.62 ± 0.30 | 2.21 ± 0.17 | 3.01 ± 0.07 |

Data reported are mean ± SD of a minimum of three determinations for each variety of juice. nd = not detected.

### 4.2. Secondary Metabolites

Secondary metabolites represent the major health-promoting benefits of sweet or­ange fruits. Polyphenols, limonoids, phenylethylamine alkaloids, carotenoids, and terpe­noids are the most represented classes [90].

Polyphenols are the principal antioxidants in the human diet [91] and are composed of several flavonoids belonging to many subclasses: flavanones, flavonols, flavones, fla­vanols, isoflavones, and anthocyanidins together with lignin, phenolic acids, and tannins. Many species-specific flavanones (i.e., hesperidin, naringin, and 12neohesperidine), fla­vones (i.e., apigenin, diosmetin, and luteolin), polymethoxyflavanones (i.e., nobiletin, sinensetin, and tangeretin), and anthocyanins (i.e., cyanidin and delphinidin derivatives) are present in Citrus fruits [53]. Although they are non-nutritive agents, citrus flavonoids exert anticancer, antimicrobial, antioxidant, anti-inflammatory, heart protection, and an­tiallergic actions [92–94]. The importance of flavonoids in sweet orange fruits has been extensively reviewed [31,90,95], and it has been ascertained that their amount and com­position greatly vary depending on variety (Table 2), maturity, the region of cultivation, and many other environmental conditions. Anthocyanins are water-soluble pigments in­volved in plant development and defense mechanisms and are contained in several culti­vars of blood oranges. It has been shown that flavonoids play a role in cancer prevention, and, particularly, the second class of citrus flavonoids (polymethoxyflavones, PMF) was identified as an important anticancer dietary factor [96,97].

Phenolic acids together with flavonoids contribute to the antioxidant activity of sweet orange fruits [98]. The sweet orange peel is a rich source of 4'-geranyloxyferulic acids, ferulic acid derivatives valued for their anticancer and anti-inflammatory activities [99]. Additionally, sweet orange seed oil contains considerable amounts of total phenolic com­pounds and can be used as a special oil in one's diet [100].

Carotenoids, responsible for the coloration of the mature fruit in most Citrus species, show a large diversity among their species and cultivars, which has a strong impact on their commercial acceptability [101]. Compared to other species, sweet oranges accumu­late larger concentrations of carotenoids together with the mandarin [56,102]. To date, ap­proximately 115 carotenoids have been identified in citrus and, among them, several are considered precursors of vitamin A [103] and are involved in the antioxidant activity of Citrus fruits [20]. In some sweet orange mutants, such as the "Cara Cara" navel orange [30], the "Kirkwood Red", the "Red Lina" [43], and the "Hong Anliu" [104], lycopene is the main carotenoid accumulated in the albedo and the juice sacs, reaching a concentration

that is 1000-fold higher than that in wild-type fruits. Lycopene-accumulating citrus mutants are attracting great interest due to their appealing red pulp color (Figure 6) and health benefits [105].

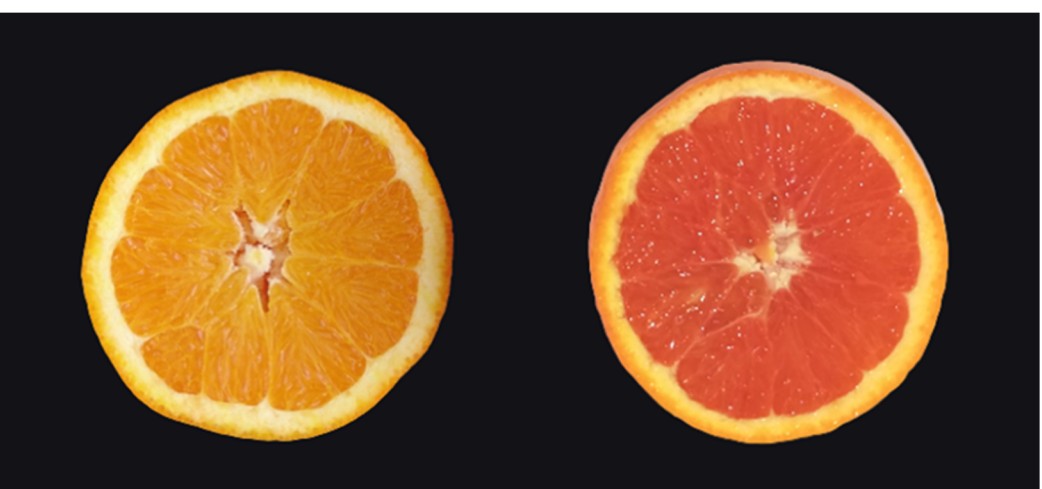

**Figure 6.** Fruit of "Cara Cara" (**right**) compared to the probable ancestor "Washington Navel" (**left**).

Limonoids are an important group of secondary metabolites composed of oxygenated triterpenoids that are present in the different fruit parts of sweet oranges [106]. Recently, particular emphasis has been given to the effects of the main biological properties of limonoids on human health. The limonoids have chemopreventive [107], antibacterial [108], and antifungal [109] activity, but they also bring an off taste, especially for the production of orange juice. The bitterness sensation from the accumulation of limonoid compounds is derived from the physical breakdown of the juice sacs, which is caused by squeezing but also, in the field, by physical damage or a freeze event. This occurrence causes the hydrolyzation of a tasteless limonoid aglycone precursor (limonate A-ring lactone) to a bitter limonoid aglycone, the limonin. Consequently, certain commercial citrus varieties, such as those belonging to the navel group, are commercialized almost exclusively as fresh fruit [110]. The bitterness caused by limonin is only detected after storage or after juice heat treatment [111]. Removing bitterness from citrus fruit juices is, to date, an important research goal. So far, the more reliable adopted methods include lye treatments, sugars addition, β-cyclodextrin and hot water treatments, other physical and chemical methods, and the use of specific microbial consortia [112–114]. However, chemical methods are costly and cannot (yet) be adopted at an industrial scale, and the potential of genetic engineering for modifying target synthetic pathways could solve this problem definitively [115].

Sweet orange fruits contain phenethylamine alkaloids, such as synephrine tyramine, N-methyltyramine, octopamine, and hordenine [116]. Synephrine alkaloid, at elevated doses, accelerates the body's metabolism and fat oxidation. Supplementation with a *C. sinensis* extract is being widely used to induce weight loss, regulating the metabolism of fatty acids [117].

The typical sweet orange aroma is due to a complex combination of several volatile organic compounds (VOCs) among which terpenoids and liposoluble terpenes are present in the flavedo (peel oil). It has been reported that the blond varieties (the "Washington navel" and "Naveline") are richer than the blood varieties (the "Moro" and "Sanguinello") in valencene among their terpenes and (E)-2-hexenol [118].

## 5. Genetic Improvement in the Sweet Orange

The sweet orange is subjected to intensive breeding programs worldwide through conventional (mostly selection and mutation breeding, but also through hybridization)

and nonconventional methods (somatic hybridization and genetic engineering). The major objectives of sweet orange improvement programs include a wide range of traits, such as tolerance or resistance to biotic and abiotic stresses, the ripening period, postharvest behavior, yield, improved fruit quality for fresh consumption (i.e., peel and flesh color, flavor, seedlessness, and beneficial compound content), and industrial transformation (i.e., juice yield, color, and TSS:TA ratio).

The conventional breeding approaches for sweet orange improvements are almost exclusively based on the selection of spontaneous or induced mutations. One of the main reasons for this limitation is related to the polyembryonic nature of the sweet orange seed combined with its high degree of heterozygosity, which favored the selection of commercial orange varieties from seedlings of nucellar origin through either spontaneous or induced mutations rather than through crossing. One exception is represented by the hybrid "Ambersweet" ((*C. clementina* × Tangelo "Orlando") × 15-3 (seedling of *C. sinensis*)), which was released in 1989 and is very similar to the *C. sinensis* varieties in terms of the chemical and organoleptic characteristics of its juice. Although it has been included among those that can be processed into "orange juice" in Florida, its low yields and poor juice quality hamper the use of this variety [119]. Another reason that limits the use of hybridization is related to the juice industry regulations, which impose worldwide quality conditions for the orange juice derived from *C. sinensis* (or the "Ambersweet" in the USA) and not from any other citrus species or hybrids (except in the USA for small quantities of the species *Citrus reticulata* and their hybrids, no more than 10% in mixture with *C. sinensis*) [120,121]. This limitation is in contrast with the recent genomic information obtained through the de novo sequencing or resequencing of several genomes, which clearly indicated that mandarin–pummelo admixtures did not differ from many type-2 or type-3 mandarins [9,122].

Clonal selection is relatively simple since it allows for the improvement of one of a few characteristics in a well-defined phenotype. However, the extremely low level of intraspecific genetic diversity exposes the cultivars to many harmful pests and diseases due to a lack of sources of resistance.

The generation of sweet-orange-like hybrids is more challenging compared to clonal selection, but it is now needed to introgress favorable genes from other citrus species or relatives. To put it into perspective, hybridization coupled with marker-assisted selection could facilitate the generation and selection of promising hybrids. However, the long juvenile period (about 5 years) and the juvenile characteristics of the first fructifications, including a tendency towards a large size, puffiness, and a thicker peel [123], generally extend the evaluation of promising new cultivars, with repercussions in the terms of the cost and time of the development of a conventional breeding program [124]. Nevertheless, in addition to the already mentioned "Ambersweet", several sweet-orange-like hybrids with the "Ambersweet" as one parent were more recently obtained (Table 3) in order to increase the HLB tolerance, with the introduction of the genes of other species considered to be more tolerant. Hybrids similar to the sweet orange in fruit size, color, and taste were selected by the USDA citrus scion breeding program and are currently under evaluation as potential cultivars [125,126]. Recently, an HLB-tolerant sweet-orange-like hybrid named the US Sun Dragon was released [127]. This hybrid has a small proportion of *Poncirus trifoliata* in its parentage and has been proposed to be used in the juice industry for the production of fruits.

**Table 3.** Sweet-orange-like hybrids from the USDA citrus scion breeding program.

| Selection | Female Parent | Male Parent |
|---|---|---|
| FF-1-64-97 | "Ambersweet" | "Tunis" sour orange × "Succory" sweet orange |
| FF-1-65-55 | "Ambersweet" | "Tunis" sour orange × "Succory" sweet orange |
| FF-1-75-55 | "Ambersweet" | "Wilking" × "Valencia" |
| FF-1-76-50 | "Ambersweet" | "Wilking" × "Valencia" |
| FF-1-76-52 | "Ambersweet" | "Wilking" × "Valencia" |

The primary method for sweet orange improvement includes the selection of spontaneous seedling or branch mutations discovered among trees growing in orchards. Indeed, all the worldwide cultivated sweet orange varieties are derived from somatic mutations that occurred in the genealogy of a single ancestor and accumulated in the different growing areas thanks to vegetative propagation [66]. The sweet orange is prone to mutations caused by somatic variation in a single cell which are transmitted between generations. This change can persist and populate a whole meristem, leading to the setting up of new variants [128]. Somatic variation is a common phenomenon in most perennials and represents a major source of genetic variability, especially for those crops that are propagated asexually through grafting. In the sweet orange, somatic variation has led to a wide range of phenotypes affecting the tree habit, juvenility, maturity date, fruit quality, and yield [129]. It has been estimated that about 80% of the sweet orange varieties cultivated worldwide arose from somatic mutants [128]. Several studies have investigated the molecular basis of the somatic mutations responsible for horticulturally important traits and cultivar diversification (Table 4). For example, the presence of a transposable element (TE) in the promoter of the *Ruby* gene was correlated to the red pigmentation in the blood orange, while TEs in the *AN1* gene were associated with fruit acidity in the "Vaniglia" sweet orange [38,130]. A number of sweet orange varieties have been resequenced, allowing the identification of different types of somatic variations, including a set of single-nucleotide polymorphisms (SNPs), insertions and deletions (InDels), and structural variations (SVs) specific to each cultivar/clone, and, thus, are useful as markers for sweet orange fingerprinting [131]. In a recent study, the sequencing of 114 somatic mutants of the sweet orange revealed an abundant set of SNPs, InDels, SVs, and transposable element (TE) insertions, the latter of which has been found to affect the genes associated with variation in fruit acidity [128].

Mutagenesis has been used worldwide to obtain seedless clones of the most important commercial seeded varieties [132]. Here, the exposure of budwoods to different doses of radiation can produce a wide range of random mutations, often resulting in diminished fertility due to a high frequency of pollen or ovule abortion [133]. For example, seedless clones were obtained through the irradiation of the "Pineapple" [134]; the "Jincheng" [135]; and, more recently, from the "Kozan" sweet orange, one of the well-known local varieties in Turkey [136]. Advances in plant cell and tissue in vitro cultures and the development of plant regeneration protocols have supported the multiplication and the propagation of novel varieties obtained through mutagenesis. Somaclonal variation indicates the genetic variation present in regenerated plants that are either uncovered or induced through a tissue culture process. The sweet orange is well suited for studies of somaclonal variation due to its nucellar embryony and its efficient performance in tissue cultures. In the USA, somaclones of the "Hamlin" and "Valencia" sweet oranges have been obtained via nucellar selection, the regeneration of adventitious shoot buds, the regeneration of a secondary embryogenic callus, and/or regeneration from a protoplast via somatic embryogenesis [45,137]. Of these, some selections were released with interesting traits linked to their season of maturity; their seed numbers; their fruit color; their flavor; and, possibly, their disease tolerance [138]. In Italy, a 30-year breeding program using

nucellar selection at CREA has produced new "Tarocco" blood orange clones, such as the "Scirè" D2062, "Meli" C8158, and "Lempso" C5787, exhibiting interesting traits linked to their ripening period and their pulp and peel pigmentation level [139]. In addition, from the genetic improvement program conducted by the University of Catania, two particularly interesting clones, among others, were selected and subsequently recovered through micrografting, the "Tarocco Ippolito" and "Tarocco Sant'Alfio"; the former is characterized by an intense coloring of the fruit as well as its excellent overall quality, and the latter is characterized by its late ripening period, which allows for the extension of the "Tarocco" harvest calendar until May [140,141].

**Table 4.** Genes/markers regulating important traits in sweet orange.

| Gene/Marker | Trait | Reference |
|---|---|---|
| *Ruby* | Anthocyanin pigmentation | [36,39] |
| *Noemi* | Anthocyanin pigmentation and fruit acidity | [38] |
| *CitPSY, CitPDS, CitZDS, CitLCYb, CitHYb, CitZEP, Csβ-LCY2,* and *CCD4b* | Carotenoid accumulation | [142–144] |
| *CsLOB1* and *CsWRKY22* | Citrus canker development | [145–148] |
| *AN1, NHX,* and *RAE1* | Fruit acidity | [128,130] |
| *CsMIPs* and *CsTALEs* | Response to biotic/abiotic stresses | [149,150] |
| *CitRWP* and *CiRKD1* with a MITE insertion | Apomixis | [151,152] |
| *VINV, CWINV1, CWINV2, SUS4, SUS5, SPS1, SPS2, VPP-1,* and *VPP-2* | Sugar accumulation in fruit juice sacs | [153] |
| SNP08 marker | Alternaria brown spot (ABS) resistance | [154] |
| *CsERF74, CsNAC25, PGs, PMEs, CCOAMTs, OMT1,* and *CAD* | Pulp tenderness | [155] |

Somatic hybridization represents another strategy for genetic improvement; somatic hybrids can be obtained through the protoplast fusion of the parental cells using electrical and/or chemical protocols. Hence, this strategy overcomes the difficulties of sexual incompatibility and represents a significant tool in ploidy manipulation [156]. Several allotetraploid somatic hybrids have been obtained from combinations of different citrus species and have been evaluated for their applications in the breeding of citrus species [157].

Advances in genomics and biotechnology strategies provide useful resources for genetics and breeding improvements in citrus species. Several molecular markers have been identified; pest and disease resistance [158–160], fruit quality [161], and polyembryony [162] are the most important traits, and linked markers are currently used for marker-assisted selection (MAS), allowing breeders to make an early selection of young progeny that exhibit the desired traits (Table 4).

The availability of the complete genome sequence of several citrus species speeds up the adoption of novel molecular-based breeding strategies. The first complete genome sequence of *C. sinensis* was released in 2013 [163] through the sequencing of the "Valencia" sweet orange. New sequencing platforms allowed the release of additional genome sequences of several sweet orange genomes (Table 5), providing a valuable resource for deciphering and manipulating traits of agronomic interest.

**Table 5.** Features of the released *Citrus sinensis* sweet orange genomes.

| Submitter | Cultivar | Sequencing Technology | Assembly Name | Assembly Level | Contig N50 (lb) | Size (Mb) | Submission Date | Bioproject | Biosample ID |
|---|---|---|---|---|---|---|---|---|---|
| China sweet orange genome project | "Valencia" | Illumina | Csi_valencia_1.0 | Chromosome | 49.9 | 327.7 | 12/12/2012 | PRJNA86123 | SAMN02981414 |
| DOE-Joint Genome Institute | "Ridge Pineapple" | 454 GS-FLX Titanium, 454 FLX Standard, and ABI 3739 | Citrus_sinensis_v1.0 | Scaffold | 6.6 | 319.2 | 30/05/2014 | PRJNA225968 | SAMN02389851 |
| Huazhong Agriculture University | "Valencia" | PacBio and Illumina GAII | ASM1810434v1 | Scaffold | 2102.1 | 338.4 | 20/04/2021 | PRJNA347609 | SAMN05893359 |
| Huazhong Agriculture University | HZAU_DHSO_2021 | Oxford Nanopore | ASM1810577v1 | Chromosome | 24,160.9 | 334.3 | 23/04/2021 | PRJNA347609 | SAMN16516428 |
| Huazhong Agriculture University | SO3 | PacBio Sequel | ASM1914366v1 | Chromosome | 246.2 | 310.6 | 06/07/2021 | PRJNA321100 | SAMN07311581 |
| Huazhong Agriculture University | TCPS1 | PacBio Sequel | ASM1914415v1 | Chromosome | 266.1 | 346.5 | 06/07/2021 | PRJNA321100 | SAMN07313349 |
| Huazhong Agriculture University | NW | Oxford Nanopore | ASM1914418v1 | Chromosome | 1932.8 | 322.6 | 06/07/2021 | PRJNA321100 | SAMN07313221 |
| Huazhong Agriculture University | NHE | PacBio Sequel | ASM1914419v1 | Chromosome | 251.3 | 315.1 | 06/07/2021 | PRJNA321100 | SAMN05412752 |
| Huazhong Agriculture University | BT2 | Oxford Nanopore | ASM1914422v1 | Chromosome | 1218.0 | 330.2 | 06/07/2021 | PRJNA321100 | SAMN07311744 |
| Huazhong Agriculture University | UKXC | Oxford Nanopore | ASM1914424v1 | Chromosome | 1693.9 | 328.7 | 06/07/2021 | PRJNA321100 | SAMN07313355 |
| Clemson University | "Valencia" | PacBio Sequel II | DVS_A1.0 | Chromosome | 32,942.3 | 299.0 | 11/02/2022 | PRJNA736174 | SAMN19611724 |
| Clemson University | "Valencia" | PacBio Sequel II | DVS_B1.0 | Chromosome | 32,342.9 | 299.6 | 11/02/2022 | PRJNA736176 | SAMN19611724 |

Genetic transformation is an efficient method for citrus genetic improvement, allowing the introgression of traits of interest into specific known genotypes and overcoming the problems related to sexual hybridization and the long juvenile phase. Genetic transformation protocols have been developed starting from many sources of explants, such as internodes, epicotyls, embryogenic cell suspensions, and protoplasts. Regeneration and transformation systems from the mature material of the sweet orange can be used in combination with the expression of the early-flowering genes to bypass the juvenile phase and to allow a rapid evaluation of modified horticultural traits [164]. Different genes are introduced into the sweet orange by using *Agrobacterium*-mediated transformation or polyethylene glycol methods to confer disease and pest resistance [165–175], fruit quality [151,176–178], and abiotic stress tolerance [179,180]. However, the transgenic approach involves the introgression of foreign DNA, and this affects the acceptability of the new products obtained and the use of these fruits for commercial purposes because they are categorized as genetically modified organisms (GMOs). New plant breeding techniques

(NPBTs), such as cisgenesis and genome editing, are expected to greatly support the genetic improvement of citrus species, overcoming the limits of conventional breeding and transgenesis. These approaches allow the introgression and/or the editing of specific desired genes into commercial varieties without altering their genetic background and without the presence of foreign DNA. Specifically, cisgenesis involves the introgression into the recipient genome of genes derived from cross-compatible species. Meanwhile, genome editing generates specific mutations in a precise position of the sequence with a low probability of inducing undesired errors and without leaving foreign DNA [181]. Among the genome editing techniques, the CRISPR/Cas9 system represents the most promising strategy. Targeted genome modification in the sweet orange using the CRISPR system with the aid of Xcc-facilitated agroinfiltration was first reported by Jia and Wang [182,183]. The CRISPR/Cas9 system has been successfully used to generate the canker-resistant "Hamlin" and "Wanjincheng" sweet oranges by targeting the CsLOB1 [145,146] and CsWRKY22 genes [147]. Recently, editing protocols have been adopted in order to produce sweet orange plantlets whose fruits contain both lycopene and anthocyanins. Five different anthocyanin-rich sweet oranges, belonging to the "Tarocco" and "Sanguigno" varieties, were transformed using the EHA105 *Agrobacterium tumefaciens* strain. This method employs a dual single-guide RNA (sgRNA)-directed genome editing approach to knock out the fruit-specific beta cyclase 2 gene that is responsible for beta-carotene biosynthesis. The obtained mutation consists of a large deletion as well as of a specific mutation in both sgRNA targets. Among the transformed plantlets, more than 80% of them were successfully edited [184].

## 6. Future Perspectives of Genetic Improvement

Despite the limits of the genetic and reproductive biology of citrus species, the conventional breeding methods still represent the major strategy for sweet orange genetic improvement. The advances in genomics and biotechnology strategies have enabled a better understanding of the genome structure and phylogenesis of most citrus species in addition to the molecular mechanisms regulating important citrus traits. The development of NGS technology has enhanced the accumulation of citrus genome sequence resources and has facilitated the release of new molecular markers [185]. The application of genome-wide association studies (GWAS) and genomic selection (GS) in citrus populations has allowed for the deciphering of the control mechanisms of different qualitative traits, such as fruit weight and peel and flesh color [186,187]. The availability of molecular markers has allowed the possibility of applying MAS, minimizing the period of trait evaluation in new selections. Transgenesis provides an efficient alternative for citrus genetic improvement, allowing the introgression of traits of interest into specific genotypes and overcoming the sexual barriers and the limits of conventional breeding. However, legal and ethical issues linked to the presence of foreign DNA that does not arise from natural events hampers its commercial utilization, limiting its use. Meanwhile, among NPBTs, the CRISPR/Cas system is now the most promising strategy for citrus genetic improvement, allowing the limits of the conventional breeding strategies and the legal issues of transgenesis to be overcome. Currently, efforts are focused on the generation of genome-modified citrus varieties via the transient expression of the CRISPR/Cas constructs. Several methods have been developed for other crops [188–191]. Recently, transfer-DNA-free base-edited citrus plants have been successfully generated by combining the use of a base editor system and the herbicide selection agent imazapyr [71,146], achieving significant improvements in the CRISPR/Cas9 system for citrus gene editing through the generation of biallelic/homozygous mutants. Additionally, new emerging delivery systems have been developed for the transient expression of the editing vector, such as the CRISPR ribonucleoprotein (RNP) complex and several nanoparticles, including nonviral carrier nanodelivery systems, which are useful for obtaining transgene-free plants with a high editing efficiency and reduced off-target effects [192]. Currently, the regulation of the NPBTs, including genome editing, is still unclear. In Europe, products resulting from NPBTs are subject

to GMO regulations according to directive 2001/18/EC. Meanwhile, the United States of America, Argentina, Brazil, Chile, Colombia, Australia, and Japan exempt plants derived from NBTs from those regulations foreseen for GMOs since they do not contain a novel combination of genetic material and could be theoretically developed through the use of conventional breeding strategies [193]. Studies with stronger scientific arguments are currently being conducted in order to review the regulation of NPBTs because they are considered to be a powerful tool contributing to sustainable agrifood systems.

**Author Contributions:** Conceptualization, S.L.M. and G.D.; writing—original draft preparation, S.S. and S.B.; writing—review and editing, M.D.G., M.C., A.G., S.L.M. and G.D. All authors have read and agreed to the published version of the manuscript.

**Funding:** This research received no external funding.

**Institutional Review Board Statement:** Not applicable.

**Acknowledgments:** This research was funded by the Ministero dell'Istruzione dell'Università e della Ricerca—DD.MM. n. 1061/2021 e 1062/2021 nell'ambito dell'Azione IV.4—'Dottorati e contratti di ricerca su tematiche dell'innovazione' del nuovo Asse IV del PON Ricerca e Innovazione 2014-2020 'Istruzione e ricerca per il recupero' D.M. n. 1062/2021 nell'ambito dell'Azione IV.4—'Dottorati e contratti di ricerca su tematiche dell'innovazione' del nuovo Asse IV del PON Ricerca e Innovazione 2014-2020 'Istruzione e ricerca per il recupero' (CUP E61821004310007 and E69J21011300006), University of Catania and AAT Oranfresh spa.

**Conflicts of Interest:** The authors declare no conflict of interest.

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
