# Peer review of "Sweet Orange: Evolution, Characterization, Varieties, and Breeding Perspectives"

_agriculture, doi:10.3390/agriculture13020264_

Round 1

Reviewer 1 Report

The work offers a complete summary of key quality characteristics regarding the 4 main Citrus sinensis varieties: (i) common oranges, (ii) Navel oranges, (iii) blood oranges and (iv) acidless oranges. Additionally, the changes in main bioactive compounds as well as in the qualitative traits depending on environmental and agronomical conditions are reviewed. Finally, a state of the art of genetic and biotechnological methods for the improvement of the genus is pointed out.

Several minor formats changes should be done in the MS:

1.    All the varieties names should be in the same format with quotation marks (‘Valencia’, ‘Hamlin, ‘Pera’, ‘Cara cara’,…) along the manuscript. Please, unified them.

2.    There are two ‘Figure 2’ in the paragraph ‘’2. One hybrid, different fruits’’. The second one ‘’ Figure 2. Collection timing of the mainly widespread varieties in a Mediterranean climate. ‘’ should be named as ‘’Figure 3.  Collection timing of the mainly widespread varieties in a Mediterranean climate,’’ as is referred along the MS.

3.    In the paragraph ‘’2.1. Common oranges’’, the line ‘‘The success of ‘Velencia’ orange has been largely…’’ should be ‘’ The success of ‘Valencia’ orange has been largely…’’.

4.    In the paragraph ‘’2.3. Pigmented or blood oranges’’ the line ‘it includes the ‘Shamouti Maouard’i and…’’ should be corrected.

5.    Related to anthocyanin accumulation during post-harvest storage, instead the reference from Latado et al 2008, please cite more recent and complete works, such as those from A. Lo Piero or L. Peña groups.

6.    All Brix percentage must be indicated as ºBrix.

Author Response

Reviewer 1: The work offers a complete summary of key quality characteristics regarding the 4 main Citrus sinensis varieties: (i) common oranges, (ii) Navel oranges, (iii) blood oranges and (iv) acidless oranges. Additionally, the changes in main bioactive compounds as well as in the qualitative traits depending on environmental and agronomical conditions are reviewed. Finally, a state of the art of genetic and biotechnological methods for the improvement of the genus is pointed out. Several minor formats changes should be done in the MS: ï‚· All the varieties names should be in the same format with quotation marks (‘Valencia’, ‘Hamlin, ‘Pera’, ‘Cara cara’,…) along the manuscript. Please, unified them. The text was modified as suggested. ï‚· There are two ‘Figure 2’ in the paragraph ‘’2. One hybrid, different fruits’’. The second one ‘’ Figure 2. Collection timing of the mainly widespread varieties in a Mediterranean climate. ‘’ should be named as ‘’Figure 3. Collection timing of the mainly widespread varieties in a Mediterranean climate,’’ as is referred along the MS. The text was modified as suggested. ï‚· In the paragraph ‘’2.1. Common oranges’’, the line ‘‘The success of ‘Velencia’ orange has been largely…’’ should be ‘’ The success of ‘Valencia’ orange has been largely…’’. The text was modified as suggested. ï‚· In the paragraph ‘’2.3. Pigmented or blood oranges’’ the line ‘it includes the ‘Shamouti Maouard’i and…’’ should be corrected. The text was modified as suggested. ï‚· Related to anthocyanin accumulation during post-harvest storage, instead the reference from Latado et al 2008, please cite more recent and complete works, such as those from A. Lo Piero or L. Peña groups. Updated ï‚· All Brix percentage must be indicated as ºBrix.
The text was modified as suggested.

Reviewer 2 Report

The review from Seminara et al, strikes an important issue for the Citrus Industry worldwide given that sweet orange is the most planted specie, and this kinds of review through an important and necessary up-to-date state of knowledge of the key steps for breeding. The Review describes profusely the main characteristics of the specie, from botanical/pomological traits, evolution, fruit quality, commercial parameters, metabolite biosynthesis and even their impact on human health. Thereafter provides a Genetic improvement perspective withs some examples of the NPBTs .

In its current form, in my opinion, the title of the review should better describe the main content of the manuscript, for example: Sweet Orange: Evolution, characterization, varieties and perspectives of breeding techniques. If authors decides to maintain the current title, the article appears to be too disperse, given few final spots of the real genetic and improvement techniques. Therefore, I would recommend shortening several paragraphs and items that are not directly tackled  by genetic improvement, thus avoiding pomological traits, profuse chemical characterization, scion/rootstocks interactions, or even commercial parameters, among others.

Besides, although the review is written by highly prestigious researchers, I noted that syntaxis could be improved by providing the review of a native English speaker. I am convinced that this  will greatly help to improve the review presentation.

Therefore, in my opinion the article should be published after the major suggested modification.

Author Response

Reviewer 2: The review from Seminara et al, strikes an important issue for the Citrus Industry worldwide given that sweet orange is the most planted specie, and this kinds of review through an important and necessary up-to-date state of knowledge of the key steps for breeding. The Review describes profusely the main characteristics of the specie, from botanical/pomological traits, evolution, fruit quality, commercial parameters, metabolite biosynthesis and even their impact on human health. Thereafter provides a Genetic improvement perspective withs some examples of the NPBTs . ï‚· In its current form, in my opinion, the title of the review should better describe the main content of the manuscript, for example: Sweet Orange: Evolution, characterization, varieties and perspectives of breeding techniques. If authors decides to maintain the current title, the article appears to be too disperse, given few final spots of the real genetic and improvement techniques. Therefore, I would recommend shortening several paragraphs and items that are not directly tackled by genetic improvement, thus avoiding pomological traits, profuse chemical characterization, scion/rootstocks interactions, or even commercial parameters, among others. The title was modified and some paragraphs were shortened. ï‚· Besides, although the review is written by highly prestigious researchers, I noted that syntaxis could be improved by providing the review of a native English speaker. I am convinced that this will greatly help to improve the review presentation. The text was revised by a native English speaker. Therefore, in my opinion the article should be published after the major suggested modification.

Reviewer 3 Report

It is a complete review article about orange breeding strategies and genetic improvement.

To make the content more practical, the genetic basis of important traits as well as MAS markers could be presented in the form of a table.

Author Response

Reviewer 3: It is a complete review article about orange breeding strategies and genetic improvement. ï‚· To make the content more practical, the genetic basis of important traits as well as MAS markers could be presented in the form of a table.
Table 4 “Genes/marker regulating important traits in sweet orange.” was included. ï‚· Concerning the comments in the attached pdf file:
- Figure 1: was included the reference as suggested.
- Figure 2: unfortunately, we don’t have photos in cross-sectional view and many varieties ripen in spring.
- What are the exact values of TSS and TA for industrial use and which variety has been improved based on this: the text was updated with “The external appearance of fruits, however, is not particularly important for industrial use although many grade standards are usually required for processing [46]. Furthermore, in the United States, some varieties (such as 'Valencia B9-65' and 'Hamlin N13-32') have been specifically selected for the improved juice characteristics [47].”
- Bone formation is mostly related to vitamin D, give references for that: https://doi.org/10.1002/jbmr.2709
-The reference N.7 was updated: “Legge, J. The Chinese Classics; BoD–Books on Demand., 1865;”

Round 2

Reviewer 2 Report

I would like to thank authors who have made sufficient effort to improve the paper presentation. Some minor suggestions and comments are included in the attached file which should be checked. After this, in my opinion the paper should be ready for publication.

Sincerely

Author Response

Authors took into account all the referee indications